# Epigenetic Regulation of Macrophage Polarization in Cardiovascular Diseases

**DOI:** 10.3390/ph16020141

**Published:** 2023-01-18

**Authors:** Sumra Komal, Sheng-Na Han, Liu-Gen Cui, Miao-Miao Zhai, Yue-Jiao Zhou, Pei Wang, Muhammad Shakeel, Li-Rong Zhang

**Affiliations:** 1Department of Pharmacology, School of Basic Medical Sciences, Zhengzhou University, Zhengzhou 450001, China; 2Jamil-ur-Rahman Center for Genome Research, Dr. Panjwani Center for Molecular Medicine and Drug Research, International Center for Chemical and Biological Sciences, University of Karachi, Karachi 75270, Pakistan

**Keywords:** epigenetics, macrophage polarization, N6-methyladenosine, non-coding RNAs, cardiovascular diseases

## Abstract

Cardiovascular diseases (CVDs) are the leading cause of hospitalization and death worldwide, especially in developing countries. The increased prevalence rate and mortality due to CVDs, despite the development of several approaches for prevention and treatment, are alarming trends in global health. Chronic inflammation and macrophage infiltration are key regulators of the initiation and progression of CVDs. Recent data suggest that epigenetic modifications, such as DNA methylation, posttranslational histone modifications, and RNA modifications, regulate cell development, DNA damage repair, apoptosis, immunity, calcium signaling, and aging in cardiomyocytes; and are involved in macrophage polarization and contribute significantly to cardiac disease development. Cardiac macrophages not only trigger damaging inflammatory responses during atherosclerotic plaque formation, myocardial injury, and heart failure but are also involved in tissue repair, remodeling, and regeneration. In this review, we summarize the key epigenetic modifications that influence macrophage polarization and contribute to the pathophysiology of CVDs, and highlight their potential for the development of advanced epigenetic therapies.

## 1. Introduction

Cardiovascular diseases (CVDs) are the leading diseases in terms of prevalence and mortality and associated with serious health and socioeconomic burden globally. Despite significant advancements in treatment and prevention, CVDs remain the major cause of death worldwide [1]. It is estimated that more than 17.9 million people die from CVDs each year [2]. CVDs are chronic, progressive diseases that irreversibly alter the myocardial architecture and ultimately lead to complications, such as arterial thrombosis and ischemic stroke [3].

Arterial hypertension, alcoholism, cholesterolemia, diabetes mellitus, obesity, and smoking are the most common risk factors associated with CVDs [4]. Furthermore, infection and inflammatory conditions increase the risk of CVDs. Severe inflammation produces a variety of complications, including atherosclerosis, viral myocarditis, and myocardial damage [5,6,7]. Recent genome-wide association studies (GWAS) and massively parallel sequencing or next-generation DNA sequencing (NGS) have provided insight into the genetic and epigenetic factors underlying CVDs [8,9,10]. In particular, epigenetic modifications refer to chemical modifications of DNA or histones that are associated with changes in gene expression [11], and have recently been linked to macrophage polarization and CVDs.

Cardiac macrophages may also contribute to cardiomyocyte-mediated inflammation and the modulation of electrical conduction in the heart [12,13]. Macrophages are heterogeneous cells found in all organ systems and play significant roles in innate and adaptive immunity, hematopoiesis, vasculogenesis, reproduction, and systemic metabolism [14]. Macrophages exhibit distinct functional phenotypes based upon their activation states [15], characterized as naïve/non-activated macrophages (M0), classically activated macrophages (M1), and alternatively activated macrophages (M2) [16]. M0 macrophages can be polarized toward pro- or anti-inflammatory phenotypes by different stimuli [17]. M1 macrophages have pro-inflammatory properties and are responsible for host defense and pathogen clearance [18]. M2 macrophages are essential for the resolution of inflammation, wound healing, and tissue repair [18,19]. M1 macrophages can be induced from M0 macrophages by lipopolysaccharides (LPS) and interferon (IFN)-γ; and M2a, M2b, and M2c can be induced from M0 by IL-4/IL-13, immune complexes/LPS/IL-1β, and IL-10/glucocorticoids/transforming growth factor (TGF)-β, respectively [17].

Cardiac macrophages are involved in diverse biological functions, including phagocytosis, antigen presentation, and immune regulation via the production of distinct cytokines and growth factors [20]. Cardiac macrophages not only trigger damaging inflammatory responses but are also involved in tissue repair and myocardial regeneration [21]. In disease, chronic inflammation modulates the macrophage response and induces a phenotypic shift leading more toward a pro-inflammatory phenotype. These changes are associated with epigenetic and transcriptional reprogramming and are modulated by epigenetic enzymes and transcription factors [22]. For example, macrophage dysregulation in atherosclerosis is associated with the complexity of the disease [23]. Recently, studies have shown that epigenetic modifications, such as DNA methylation, histone modifications, and RNA regulation, are significantly involved in the differential activation of macrophages and contribute to macrophage polarization [24], thereby serving as potential therapeutic targets for the treatment of various CVDs [25]. In this review, we summarize the epigenetic regulatory mechanisms during macrophage polarization and their roles in the development and management of CVDs.

## 2. Macrophage Heterogeneity and Functions

Macrophage polarization is a unique phenotypic phenomenon where macrophages exhibit a particular functional response to the microenvironment [26]. Macrophage activation produces distinct functional phenotypes that maintain homeostasis primarily by modulating the release of pro-and anti-inflammatory cytokines [27]. The M1 macrophage phenotype is activated by granulocyte-macrophage colony-stimulating factor (GM-CSF), and toll-like receptor (TLR) or IL-1R ligands and secretes pro-inflammatory cytokines, such as interleukin (IL)-1β, IL-6, IL-12, IL-23 [28,29], tumor necrosis factor α (TNF-α) [30], and reactive oxygen intermediates [31]. Furthermore, they express specific biomarkers, including CD86, CD83, CD80, CD68, CD40, and major histocompatibility complex class I (MHC-I) [32]. M2 macrophages produce anti-inflammatory cytokines, including IL-10, IL-4, TGF-β, and arginase-1 (Arg-1); and exhibit elevated expression of CD206, CD204, and CD163 on the cell surface [33]. An imbalance between M1 and M2 macrophage populations is associated with left ventricle (LV) remodeling and heart failure (HF) [34]. Cardiac macrophages maintain a homeostatic population owing to their self-proliferative properties and are independent of monocyte-derived macrophages in the blood [35]. Studies have shown that the heart exhibits a distinct subset of macrophages that can be differentiated by the cell surface expression of C-C chemokine receptor type 2 (CCR2). The presence or absence of CCR2 is considered a robust marker of macrophage origin and phenotype. Further, CCR2 expression distinguishes monocyte-derived cardiac macrophages from those that are embryonic in origin. CCR2^+^ and CCR^−^ macrophage subsets exhibit distinct functions and gene expression profiles [36]. Cardiac CCR2^+^ macrophages originate from bone marrow-derived monocytes and are involved in immune surveillance, neutrophil recruitment, inflammatory cytokine production, and adverse myocardial remodeling, whereas CCR2^−^ macrophages originate from fetal monocyte progenitors and the primitive yolk sac and are involved in the clearance of apoptotic cells, production of anti-inflammatory cytokines, angiogenesis, and cardiomyocyte proliferation [20,37]. The distinct sets of CCR2^+^ and CCR^−^ tissue-resident macrophages have been reported in the human myocardium (Figure 1). In the adult heart, two resident cardiac macrophage subsets (MHC-IIlowCCR2^−^ and MHC-IIhighCCR2^−^), a monocyte-derived macrophage population (MHC-IIhiCCR2^+^) and a monocyte population (MHC-IIloCCR2^+^), have been identified by a combination of flow cytometry and genetic lineage tracing techniques; the injured adult heart selectively recruits monocytes and MHC-IIhiCCR2^+^ monocyte-derived macrophages [20]. Furthermore, cardiac macrophages facilitate electrical conduction in the heart, and their depletion can exacerbate myocardial remodeling and dysfunction, highlighting the role of resident cardiac macrophages in the pathophysiology of CVDs [21].

## 3. Role of Macrophage Polarization in the Pathophysiology of CVDs

Macrophage polarization is the key component of wound healing and tissue regeneration [38]. Monocyte-derived cardiac macrophages are actively involved in the initiation, progression, and resolution of inflammation in myocardial injuries [39]. Studies have shown that the secretion of pro-inflammatory exosomes and microRNAs (miRNAs) by M1 macrophages contribute to myocardial damage by inhibiting angiogenesis and heart repair [40]. In mice, monocytes can be divided by variable expression of lymphocyte antigen 6 complex, locus C (Ly6C). Ly6C^high^ monocytes express a high level of CCR2 and low level of CX3C chemokine receptor 1 (CX3CR1) and perform pro-inflammatory functions. Ly6C^low^ monocytes, also known as patrolling cells, express high levels of CX3CR1 and low levels of CCR2, and can promote tissue remodeling and angiogenesis [41]. Further, Ly6C^low^ monocyte recruitment to the infarcted myocardium regulates the production of M2 macrophages, reduces inflammation via the secretion of IL-10, and initiates ECM remodeling and angiogenesis [42]. In a mouse model of MI, the depletion of cardiac macrophages increases infarct size, impairs LV systolic function, and exacerbates LV remodeling [43]. In the early stages of myocardial reperfusion, M1 macrophages are thought to cause myocardial damage by producing inflammatory mediators, reactive oxygen species (ROS), and proteases [44]. Macrophage-mediated immune response plays a vital role in myocardial ischemia/reperfusion (I/R) injury. During I/R injury, macrophages show functional heterogeneity, with initial M1 macrophage infiltration, followed by M2 macrophages. M1 macrophages induce strong pro-inflammatory reactions and contribute to myocardial I/R injury. Dectin-1 expressed on cardiac macrophages and is essential for the regulation of immune homeostasis as a pattern recognition receptor. Further, it induces macrophage polarization toward the M1 phenotype, which exacerbates myocardial I/R damage [45]. Studies have demonstrated that M2 macrophages play a crucial role in alleviating myocardial I/R injury by suppressing the release of pro-inflammatory cytokines and increasing levels of anti-inflammatory cytokines, such as IL-10 [46]. Macrophages have a potential role in ventricular arrhythmias by producing pro-inflammatory cytokines, sprouting sympathetic nerves and directly influencing cardiac electrophysiology [13]. Cardiac macrophages are abundant at the atrioventricular (AV) node and facilitate electrical conduction in the steady-state heart. Moreover, cardiac macrophages electrically couple to cardiomyocytes via gap junctions containing Cx43 and assist normal AV conduction. However, the depletion of Cx43 in macrophages and any disruption in the level of Cx43 results in the blockage of conduction velocity and leads to cardiac arrhythmias. Macrophages also have a critical role in human ischemic and idiopathic dilated cardiomyopathy, as well as in heart failure with preserved ejection fraction (HFpEF) by releasing inflammatory cytokines [13,47,48]. Overall, these studies suggest that macrophages play a critical role in cardiac pathophysiology.

## 4. Epigenetic Regulation of Macrophage Polarization and CVDs

Epigenetic processes mediate the diversity of gene expression patterns in different cells and tissues [49]. Epigenetic regulation controls a variety of cellular activities, including macrophage polarization [24]. These modifications include DNA methylation, histone modifications, and RNA modifications and represent a molecular framework through which the environment modulates gene expression (Table 1) [10]. All of these epigenetic changes have potential roles in the activation and functional differentiation of macrophages (Figure 2) [50].

### 4.1. DNA Methylation

DNA methylation is an epigenetic mechanism in which DNA methyltransferases (DNMTs) covalently transfer a methyl group from S-adenyl methionine (SAM) to the C-5 position of the cytosine residue to form 5-methylcytosine (5mC) [51]. DNA methylation regulates gene expression by recruiting proteins involved in gene repression or inhibiting the binding of transcription factors to DNA, leading to the modulation of cellular activities [52]. DNA methylation is fundamental for normal development and involved in several key processes, such as X-chromosome inactivation, genomic imprinting, and the suppression of repetitive element transcription and transposition; its dysregulation can lead to diseases, such as cancer and CVDs [53]. Studies have demonstrated potential associations of DNA methylation with cardiovascular aging and disorders [54]. Smoking, high plasma homocysteine, and low levels of dietary folate are the most common cardiovascular risk factors associated with dysregulated DNA methylation [55]. DNA methylation regulates macrophage gene expression in the pathogenesis of several diseases, including inflammatory diseases [24].

DNA methylation has also been associated with monocyte-to-macrophage differentiation and subsequent macrophage activation. DNMT3B is the only known DNMT involved in M2 differentiation and phenotypic regulation [24]. Studies have shown that homocysteine (Hcy) affects macrophage subtype polarization both in vitro and in vivo, and inhibits cystathionine γ-lyase (CSE) expression level and hydrogen sulfide (H_2_S) production in macrophages, accompanied by increases in DNMT expression and DNA hypermethylation in the CSE promoter region [56]. Additionally, dysregulation of the CSE-H2S signaling pathway, which is involved in monocyte/macrophages-mediated inflammation production, contributes to atherosclerosis pathogenesis (Figure 3) [57,58]. In apolipoprotein E (ApoE)-knockout (KO) mice fed an atherogenic diet, as well as in individuals with atherosclerosis, reduced peroxisome proliferator-activated receptor gamma (PPAR-γ) and elevated DNMT1 levels in macrophages, as well as increased pro-inflammatory cytokine production, are associated with the progression of atherosclerosis [59]. It has been reported that patients with vascular diseases show considerably reduced levels of genomic DNA methylation and S-adenosylmethionine/S-adenosylhomocysteine ratios compared with those of controls. Moreover, a decrease in DNA hydroxymethylation has been observed in an animal model of cardiac hypertrophy [60]. A recent study has demonstrated that the gene-specific promoter DNA methylation status of macrophage polarization markers differs between patients with coronary artery disease (CAD) and healthy controls. A significant difference was observed between the percentage methylation of IL12b, STAT1, MHC2, JAK1, JAK2, and iNOS in CAD patients and control subjects. These findings suggest that balancing M1/M2 polarization during atherosclerosis could be a potential method to lower the risk of CVDs [61]. Furthermore, epigenetic markers including gene-specific promoter DNA methylation based on monocyte/macrophages, might serve as clinically useful diagnostic markers and therapeutic targets [62].

### 4.2. Histone Modifications

Histones are the proteins that provide structural support for a chromosome and play a crucial role in the transcriptional activities of the gene. Histones undergo a variety of posttranslational modifications, such as acetylation, methylation, phosphorylation, sumoylation, and ubiquitination, which alter the chromatin structure and subsequently affect gene expression [63]. Histone modifications are catalyzed by enzymes that primarily act on the N-terminal tail of histone proteins, mostly on arginine, lysine, serine, threonine, and tyrosine residues [64]. Histone methylation at lysine residues has an essential role in M2 macrophage activation. Histone modifications contribute to the regulation of macrophages and any changes in histone-modifying enzymes in macrophages have a significant impact on their inflammatory repertoire [65]. Enhanced histone acetylation and dimethylation of lysine 4 on histone H3 are linked to increased atrial natriuretic peptide (ANP) and B-type natriuretic peptide (BNP) gene expression in the LV [66].

Studies have shown that histone modifications, including histone methylation and acetylation, are associated with the progression and development of atherosclerosis in patients with carotid artery stenosis. The study revealed the increased histone acetylation on H3K27and H3K9, and significant changes in the methylation state of H3K4, H3K9, and H3K27 in smooth muscle cells and macrophages in advanced atherosclerotic lesions compared to healthy vessels [67]. In a murine model of atherosclerosis, the histone methyltransferase enhancer of zeste homolog 2 (EZH-2), which trimethylates H3K27, was found to aggravate atherosclerosis by suppressing macrophage cholesterol efflux via ABCA-1 (ATP-binding cassette transporter) and accelerating foam cell development [68]. In contrast, alterations in histone deacetylases (HDAC), such as Sirtuin 6 (SIRT-6) and SIRT-1, boost cholesterol efflux by activating the ATP binding cassette subfamily A member 1 (ABCA-1) and ATP binding cassette subfamily G member (ABCG-1), resulting in a decrease in macrophage-derived foam cell production and inflammation (Figure 4) [69]. Furthermore, the depletion of H3K27 demethylase exacerbated atherosclerotic plaque formation and downregulated the expression of gene sets involved in M2 macrophage polarization [70].

Histone deacetylase 3 (HDAC3) mediates macrophage polarization and is essential for the induction of the LPS-induced inflammatory response in macrophages. HDAC3-deficient macrophages have been found to be hyperresponsive to IL-4-induced polarization. Macrophage HDAC3 is potentially involved in atherosclerotic plaque development and its deletion in macrophages results in increases in plaque size, stability, and collagen deposition in HDAC3 KO mice [71]. Further, HDAC3 is associated with inflammatory macrophages and is upregulated in ruptured human atherosclerotic plaques; its levels are inversely correlated with pro-fibrotic TGFB1 gene expression. HDAC3 knockdown in macrophages reduces the innate inflammatory response triggered by LPS and decreases the production of pro-inflammatory mediators, such as TNF-α, iNOS, and IL-12. Therefore, any alteration in the macrophage epigenetic landscape may influence atherosclerosis disease outcomes [71].

Recent data support the role of histone acetyltransferases (HAT) in mediating NADPH oxidase 5 (Nox5) expression levels in human macrophages under inflammatory conditions and provide evidence that dysregulated histone acetylation-related epigenetic mechanisms contribute to ROS overproduction in atherosclerosis by the upregulation of NOX5 [72]. Further, studies have shown a correlation between histone deacetylase 9 (HDAC9), matrix metalloproteinase 12 (MMP12), and macrophage polarization in patients with carotid plaques. Dysregulated HDAC9 expression is associated with macrophage differentiation towards the macrophage subtype by MMP12 and may lead to atherosclerosis [73]. Studies show that histone modifications are associated with macrophage polarization followed by myocardial dysfunction and can be used as potential diagnostic and therapeutic targets for certain inflammatory and autoimmune diseases [74].

### 4.3. RNA Regulation

Studies have identified various types of RNA modifications in protein-coding RNAs and non-coding RNAs (ncRNAs) [75]. These RNA modifications play an important role in many biological processes, including growth, development, and cellular functions. Additionally, these modifications have a significant influence on a variety of molecular processes, such as pre-mRNA splicing, transcription, nuclear export, mRNA translation, and degradation, and consequently impact downstream transcriptional processes or gene expression [10]. RNA transcript modifications are usually determined by the coordinated action of three effector proteins, including RNA-modifying enzymes (writer proteins), YT521-B homology (YTH) domain protein family (reader proteins), and demethylases (eraser proteins) [76]. Studies show that regulation of mRNA stability through RNA modifications plays a pivotal role in the regulation of gene expression as well as in encoding pro-inflammatory and anti-inflammatory factors [10]. For example, RNA-binding proteins CPEB4 (cytoplasmic polyadenylation element binding protein 4) and TTP (tristetraprolin) act oppositely to regulate RNA stability in macrophages and modulate inflammation [77]. mRNA-based modifications involve the attachment of a methyl group at a particular position either on a bases, ribose sugar or on both base and sugar. Over 150 modifications have been reported to date, including N6-methyladenosine (m^6^A), 5-methylcytosine (m^5^C), and N1-methyladenosine (m^1^A) [78].

m^6^A RNA methylation is the most abundant posttranscriptional mRNA modification [10]. m^6^A RNA methylation is regulated by methyltransferases (methyltransferase-like [METTL] 3, METTL14, METTL16, and WT1 associated protein (WTAP)), YT521-B homology or YTH domain proteins (YTHDF1, YTHDF2, and YTHDF3), and demethylases (fat mass and obesity-associated protein (FTO) and ALKB homolog (ALKBH5)) and can affect various biological processes, including mRNA processing or stability and gene expression [79]. Recently, a study has revealed the upregulation of METTL3 and METTL14 mRNA levels via the METTL3-STAT1 axis in M1 mouse bone marrow-derived macrophages compared with levels in M2 macrophages. Further, METTL3 and METTL14 KO inhibits M1 but not M2 macrophage polarization. Thus, these loci serve as potential anti-inflammatory targets [80,81].

Studies have demonstrated that the m^6^A demethylase FTO can regulate macrophage polarization via the NF-κB signaling pathway and reduce mRNA stability and expression of PPAR-γ and STAT1 via YTHDF2-mediated degradation. FTO KO inhibits the polarization of M1 and M2 macrophages simultaneously [82]. Additionally, a study of apolipoprotein E-deficient (ApoE^−/−^) mice with adeno-associated virus serotype 9-FTO revealed a significant reduction in plasma levels of total cholesterol and low-density lipoprotein cholesterol in macrophages, thereby slowing atherosclerotic plaque formation. Hence, FTO exhibits antiatherogenic effect on foam macrophages and may serve as a potential therapeutic target for the treatment of various CVDs [83]. The m^5^C post-transcriptional modification denotes the methylation of the cytosine base at position 5 by RNA methyltransferases (e.g., NOP2/Sun domain family member 2 or Nsun2) and DNMT2 [84]. It has been reported that Nsun2 methylation upregulates the mRNA expression of Intercellular Adhesion Molecule 1 (*ICAM-1*) and regulates vascular endothelial inflammation by inhibiting M2 macrophage polarization. Alternatively, the downregulation of Nsun2 in vascular smooth muscle cells and endothelial cells protects against severe inflammatory reactions and atherosclerotic lesion formation [85]. Studies have shown that m^5^C methylation in the mRNA coding region regulates protein synthesis as well as enhances interleukin (IL)-17A expression level in plasma, macrophages, B cells, and T cells, which indirectly contributes to the onset of inflammation and atherosclerosis [86]. Therefore, it is believed that m^5^C methylation can be used as a new target in the field of mRNA-based therapeutics. Recently, a significant association between m^1^A “reader” YTHDF3 and macrophage polarization in the progression of aortic inflammation has been reported. YTHDF3 KO in macrophages significantly inhibits M1 polarization, facilitates M2 polarization, and attenuates vascular inflammation by downregulating the expression of inflammatory cytokines (e.g., TNFα and IL-1β) and upregulating the secretion of anti-inflammatory chemokines and cytokines (e.g., TGFβ and IL-10) [87].

### 4.4. Noncoding RNA Regulation

Genome-wide analyses of epigenetic and transcriptional modifications have demonstrated dynamic changes in loci associated with macrophage polarization, resulting in the regulation of distinct transcription factors and signaling pathways [88]. The non-coding genome are involved in genetic programming and gene regulation in both healthy and CVD state [89]. ncRNAs, especially miRNAs, long non-coding RNAs (lncRNAs), and circular RNAs (circRNAs), are actively involved in the regulation of the polarization of macrophages [90,91,92]. Macrophages play a crucial role in atherosclerosis as well as acute and chronic HF. In response to external stimuli, such as LPS, interferon regulatory factor 1 (IRF1), and IL-4, ncRNAs may alter macrophage polarization, gene expression, and signaling pathway activity and may contribute to the development of various CVDs, including atherosclerosis, myocardial infarction (MI), I/R injury, and HF (Table 2) [93,94].

#### 4.4.1. miRNA-Mediated Macrophage Polarization and CVDs

miRNAs are short ncRNAs (approximately 22 nucleotides) predominantly involved in posttranscriptional gene regulation. miRNAs play a critical role in various pathophysiological processes, including cell proliferation, differentiation, apoptosis, and metabolism [95]. They can regulate macrophage polarization and influence cardiovascular function [90]. Studies have shown that M1 macrophages contribute to the development of myocardial injury by secreting pro-inflammatory miRNAs and exosomes, thereby inhibiting myocardial healing and angiogenesis [96]. Numerous studies have examined the roles of miRNAs in atherosclerosis [97]. Recently, a study has revealed novel miRNA-mediated mechanisms underlying macrophage regulation and their atherogenic and atheroprotective effects on atherosclerosis. In macrophages, miR-133b-3p modulates the expression of mastermind-like protein 1 (MAML1), which could serve as a proatherosclerotic factor, followed by the regulation of the NOTCH signaling pathway, which contributes to the plaque leukocyte influx and atherosclerosis progression [98].

Additionally, the level of miR-17-5p was significantly upregulated in macrophages of an atherosclerosis murine model. However, miR-17-5p downregulation by an miR-17-5p antagomir significantly reduces the severity of atherosclerosis by reducing the production of inflammatory cytokines, upregulating ATP-binding cassette transporterA1 (ABCA1), reducing lipid accumulation, and activating the PPAR-γ/liver X receptor (LXR) α signaling pathway [99]. The downregulation of miR-378a-3p in hyperlipidemic ApoE^−/−^ mouse aortic macrophages results in a reduction of signal regulatory protein (SIRP) α and hinders the clearance of apoptotic cells within plaque lesions, accelerating the progression of atherosclerosis. The dysregulation of the CD47-SIRPα axis may also contribute to the pathogenesis of atherosclerosis in humans [100]. Additionally, miR-let7 has been shown to attenuate the progression of atherosclerosis in ApoE^−/−^ mice and facilitate M2 macrophage infiltration and polarization [101]. miR-205-5p is a regulator of macrophage lipid transport and contributes to atherosclerosis [102]. Therefore, further investigations are needed to clarify the precise role of miRNA–macrophage interactions during atherosclerosis.

Recently, a study has demonstrated the critical role of miR-21-5p in limiting the activation of pro-inflammatory macrophages in the heart after MI. Additionally, the expression of miR-21-5p in macrophages restricts the production of inflammation cytokines by targeting kelch repeat and BTB domain containing 7 (KBTBD7), a transcriptional activator associated with the mitogen-activated protein kinase (MAPK) signaling pathway. However, miR-21-5p dysregulation results in KBTBD7 overexpression and enhanced macrophage activation via the p38/NF-kB pathway [103]. Further, miR-375-3p in macrophages downregulates the production of pro-inflammatory cytokines by regulating the 3- phosphoinositide-dependent protein kinase 1 (PDK1) pathway and can augment M2 macrophage polarization. Additionally, increased expression of NADPH oxidase 2 (NOX2) has been reported in cardiomyocytes and macrophages after MI, which promotes macrophage polarization and the production of inflammatory cytokines [104]. Additionally, miR-204-5p, miR-148b-3p, and miR-106b-5p can suppress NOX2 and significantly improve infarct size and myocardial function post-MI in mice, providing potential therapeutic targets [105]. Several studies have investigated the role of macrophages in myocardial I/R injury [90,91]. A recent study has demonstrated that mesenchymal stromal cell-derived exosomes with abundant miR-182 attenuate myocardial I/R injury by shifting macrophages toward M2 over the M1 phenotype within the heart via the TLR4/NF-κB/PI3K/Akt signaling pathway [106]. Further, evidence from coxsackievirus-B3 (CVB3) murine models of viral myocarditis strongly suggests that the upregulation of miR-155-5p is involved in the dysfunctional activation of T cells during acute myocarditis as well as inflammatory monocyte/macrophage infiltration and M1 cardiac macrophages-polarization via the TLR signaling pathway [107]. Moreover, increases in the expression levels of miR-21-5p, miR-155-5p, and miR-146b-5p were observed in ventricular septal biopsies of patients with acute viral myocarditis [108]. Studies further suggest that miR-155 expression in macrophages promotes myocardial inflammation, cardiac hypertrophy, and HF in response to pressure overload. However, the inhibition of miR-155 prevents cardiac hypertrophy and HF and may serve as a suitable therapeutic target in myocardial abnormalities [109]. Furthermore, an upregulated level of miR-223 protects the murine heart from CVB3-induced myocardial injuries by regulating macrophage polarization via PBX/knotted 1 homeobox 1 (Pknox1). Therefore, a deep understanding of miRNA-mediated macrophage polarization mechanisms could provide potential targets for the management of various CVDs [110].

#### 4.4.2. lncRNA-Mediated Macrophage Polarization and CVDs

lncRNAs are non-protein-coding RNA molecules that consist of more than 200 nucleotides [111]. They modulate gene expression at both transcriptional and post-transcriptional levels and play an important role in cardiac development and the pathogenesis of CVDs [112]. Recent data suggest that lncRNAs play a crucial role in macrophage polarization, function, signaling, and diseases, and circulating lncRNAs can be used as independent biomarkers in CVD [91]. The lncRNAs TCONS-00019715 and myocardial infarction-associated transcript 2 (Mirt2) are associated with macrophage phenotypic switching. Further, Mirt2 KO repressed pro-inflammatory cytokine regulation in LPS-stimulated macrophages by attenuating NF-κB signaling. The lncRNA metastasis-associated lung adenocarcinoma transcript 1 (MALAT1) is abundantly expressed in THP-1-derived macrophages and contributes to CD36-mediated lipid uptake by macrophages. Studies have demonstrated that the macrophage-mediated lipid uptake of oxidized low-density lipoproteins plays a critical role in atherosclerotic plaque development by regulating MALAT1 transcription via the NF-κB signaling pathway [113]. Moreover, the lncRNA myocardial-infarction associated transcript (MIAT) is highly expressed in human carotid plaques and may act as a diagnostic marker in ischemic stroke [114,115].

Studies have shown that lncRNAs regulate the formation of macrophage-derived foam cells by the accumulation of lipoprotein cholesteryl esters in lysosomes and endoplasmic reticulum, followed by the formation of lipid droplets in macrophages, which ultimately contributes to the pathogenesis of atherosclerosis. Both M1 and M2 macrophages have been reported in atherosclerotic lesions [116]. Competing endogenous lncRNA 1 for miR-4707-5p and miR-4767 (CERNA1) can also stabilize atherosclerotic plaques by regulating apoptosis inhibitor 5 (API5) expression in the vascular smooth muscle cells and macrophages in ApoE^−/−^ mice [117]. Additionally, a significant increase in the expression of a lncRNA associated with the progression and intervention of atherosclerosis (RAPIA) has been reported in atherosclerotic sites in macrophages and is associated with the progression of atherosclerosis. However, the downregulation of RAPIA exerts atheroprotective effects in an ApoE^−/−^ murine model by reducing plaque size, attenuating lipid accumulation, improving the collagen content, and reducing macrophage accumulation in atherosclerotic plaques. Therefore, suppressing RAPIA expression could be a suitable therapeutic target for advanced atherosclerotic lesions [118]. Further, the lncRNA non-coding repressor of NFAT (NRON) regulates M2 polarization and improves atrial fibrosis by suppressing pro-inflammatory macrophages in cardiac myocytes [119].

In contrast, the lncRNA AK085865 can modulate macrophage polarization, and its silencing reduces M2 polarization while increasing M1 polarization. Thus, the KO of AK085865 aggravates viral myocarditis in animal models, promotes M1 polarization, and restricts M2 polarization. This indicates that AK085865 has a potential role in the regulation of macrophage polarization and may serve as a target for the management of viral myocarditis [120]. The downregulation of the lncRNA maternally expressed 3 (MEG3) in a murine model of CVB3-induced viral myocarditis reduces M1 macrophage polarization and elevates M2 macrophage polarization via the miR-223/TRAF6/NF-κB pathway, thus relieving myocarditis [121]. These studies collectively indicate that lncRNAs might serve as biomarkers of heart disease.

#### 4.4.3. circRNA-Mediated Macrophage Polarization and CVDs

circRNAs are a group of single-stranded, non-coding RNAs with diverse roles in gene regulation and cellular activities [122]. A recent GWAS has identified a nonprotein coding region on chromosome 9p21 that is strongly associated with CVDs, including MI, atherosclerosis, myocardial fibrosis, and HF [123]. The differential expression of circRNAs in bone marrow-derived macrophages among polarization stages has been demonstrated by a high-throughput circRNA microarray assay; levels of circRNA-010231, circRNA-010056, and circRNA-003780 were significantly upregulated and levels of circRNA-003424, circRNA-018127, circRNA-013630, and circRNA-001489 were downregulated in M1 macrophages and M2 macrophages, respectively [92]. Additionally, the overexpression of circCdyl promotes M1 polarization and the M1-mediated inflammatory response, both in vivo and in vitro, modifies IRF4/CCAAT/enhancer-binding protein (C/EBP)-δ expression, and substantially induces vascular inflammation, whereas a circCdyl deficiency has the reverse effects [124]. The data shows the promising role of circRNAs in promoting macrophage polarization and as a potential target for treating macrophage polarization-related diseases.

## 5. Other Mechanisms Underlying Macrophage Activation in CVDs

In macrophages, transcription factors determine the expression of specific genes and are regulated by a variety of signaling molecules [125]. Several studies have shown that subtype-specific signaling pathways and downstream transcription factors are key mediators of macrophage polarization [126]. For example, signal transducers and activators of transcription family (STAT)1, kruppel-like factors (KLFs) 6, IRF9, CCAAT/enhancer binding protein (C/EBP)-δ, (C/EBPδ), and nuclear transcription factor-κB (NF-κB) are important transcription factors involved in M1 macrophage polarization [127], whereas STAT3, STAT6, PPARs, C/EBP-β, KLF4, IRF4, c-MYC, and GATA binding protein 3 (GATA3) are associated with M2 macrophage polarization [128]. Studies have shown that members of the STAT protein family are highly expressed in cardiac myocytes, fibroblasts, and endothelial cells and regulate macrophage M1/M2 polarization. In response to various stimuli, such as hypoxia, MI, and I/R injury, the JAK/STAT signaling pathway is activated in the heart, triggering an inflammatory response in numerous CVDs, including atherosclerosis. KLFs are DNA-binding transcriptional regulators and are actively involved in the regulation of macrophage polarization. Studies have demonstrated the central role of the KLF family in the pathogenesis of various CVDs, including atherosclerosis, MI, and stroke [129]. IRFs and NF-κB are important mediators of M1/M2 polarization and are actively involved in the development of metabolic diseases, inflammatory diseases, and CVDs [130]. Accordingly, establishing a better understanding of the molecular mechanisms driving macrophage biology could lead to the discovery of new therapeutic targets for CVDs.

## 6. Macrophage Epigenetics as a Potential Therapeutic Target for CVDs

Epigenetics plays a key role in the onset and progression of several CVDs, including aortic aneurysm, cardiac hypertrophy, HF, ischemic heart disease, pulmonary hypertension, and vascular calcification, by influencing gene expression and cellular function [131]. Epigenetic mechanisms also regulate macrophage polarization and may be altered by pharmacological inhibitors and modulators, making them potential therapeutic targets for several diseases. Accordingly, research on DNA methylation may present an opportunity for the development of new therapeutics in the form of “epigenetic medicines”, such as DNMT inhibitors, and diagnostic biomarkers [132]. Although monocytes and macrophages both play a role in atherosclerosis, epigenetic indicators based on monocyte/macrophage gene-specific promoter DNA methylation could be used as diagnostic markers or therapeutic targets in clinical trials [133]. Despite numerous studies aimed at the identification of clinical epigenetic biomarkers and drugs, more work is needed to fully understand the therapeutic potential of drugs that target specific epigenetic mechanisms in monocytes/macrophages as well as M1/M2 macrophage polarization during atherosclerosis. As M1/M2 polarization is a key element in plaque stability, shifting the M1/M2 macrophage balance to a more favorable phenotype with new drugs may aid in the prevention of atherosclerosis. As a result, epigenetic gene regulation via DNA methylation-related therapies may open up new approaches for the treatment of atherosclerosis. HDACs are upregulated in aortic smooth muscle cells under mitogenic stimulation; the inhibition of HDACs leads to a reduction in smooth muscle cell proliferation, indicating atheroprotective effects [134].

Furthermore, TET-mediated DNA hydroxymethylation regulates the expression of key cardiac genes, such as myosin heavy chain 7 (Myh7), according to genome-wide mapping in embryonic, neonatal, adult, and hypertrophic mouse cardiomyocytes, implying that this epigenetic mechanism contributes to heart development and disease [60]. Data from recent years indicate that ncRNAs play a significant role in macrophage polarization and the development of CVDs and may serve as novel therapeutic targets for CVDs.

## 7. Concluding Remarks and Future Prospects

CVDs, owing to their high mortality and morbidity rates, are a substantial burden. Environmental factors and individual lifestyles contribute to disease risk; however, the association between these external risk factors and genetic factors has remained unclear. Epigenetics has significantly improved our understanding of the contributions of gene–environment interactions. Epigenetic mechanisms, such as DNA methylation, histone modification, and RNA regulation, may influence gene expression levels and interfere with cellular differentiation, leading to impaired myocardial function. Epigenetic mechanisms play a pivotal role in signal modulation and transmission during macrophage polarization and reprogramming. A deep understanding of the epigenetic regulation of macrophage phenotypes would pave the way for the development of gene-specific therapeutic strategies to augment host defense while protecting myocardial integrity and preventing the progression of CVDs.

## Figures and Tables

**Figure 1 pharmaceuticals-16-00141-f001:**
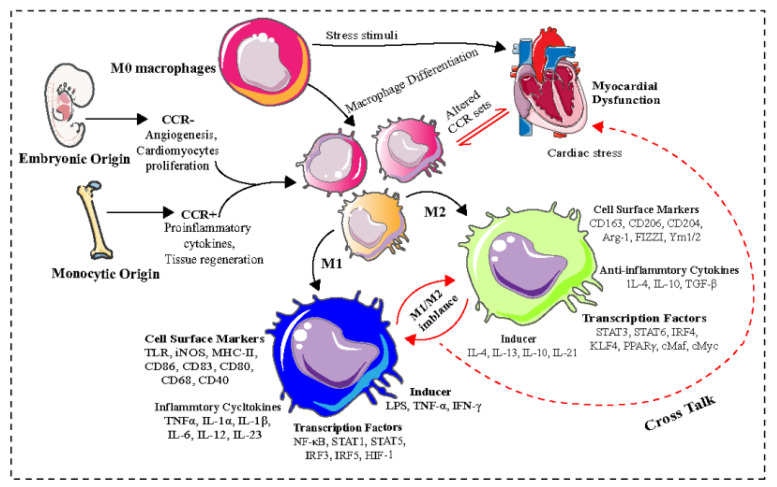
Macrophage polarization and its association with myocardial dysfunction. Macrophage polarization is a unique phenotypic expression wherein macrophages exhibit a particular functional response to the host immune system in both healthy and pathological conditions. Macrophages are also involved in triggering an inflammatory response, immune control, and adaptive immune response; whereas the imbalance between M1/M2 macrophage populations has been reported to be associated with ventricle remodeling and myocardial dysfunction. Abbreviations: CCR, C-C chemokine receptor type; M0, naïve/non-activated macrophages; LPS, Lipopolysaccharides; TNF-α, Tumor necrosis factor alpha; IFN-γ, Interferon gamma; TLR, Toll-like receptors, iNOS, Inducible nitric oxide synthase; MHC, Histocompatibility complex; CD, Cluster of differentiation; IL, Interleukin; Arg-1, Arginase 1; FIZZ1, Resistin-like molecule alpha1; TGF-β, Transforming growth factor beta; M1/M2, Macrophages; NF-κB, Nuclear factor kappa-light-chain-enhancer of activated B-cells; STAT, Signal transducer and activator of transcription; IRF, IFN regulatory factor; HIF-1, Hypoxia-inducible factor 1; KLF4, Krüppel-like factor 4; PPAR, Peroxisome proliferator- activated receptor; cMaf, transcription factor c-Maf; cMyc, c-Myc multifunctional transcription factor.

**Figure 2 pharmaceuticals-16-00141-f002:**
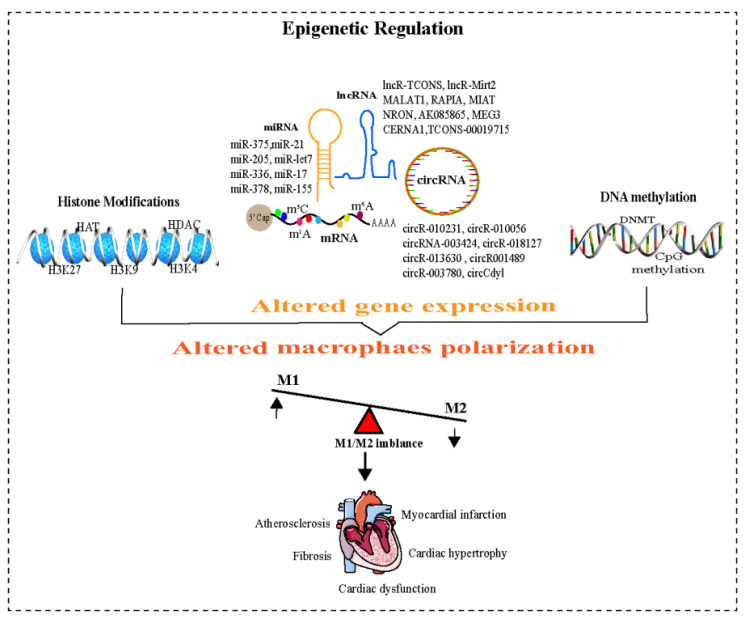
Epigenetic regulation and M1/M2 imbalance. Epigenetic regulations are associated with altered gene expression and control a variety of cellular activities, including macrophage polarization. These modifications include DNA methylation, histones modification, and non-coding RNAs (ncRNAs), which represent a molecular framework through which the environment modulates gene expression and contributes to cardiovascular diseases. Abbreviations: DNMT, DNA methyltransferases; HAT, histone acetyltransferase; HDAC, histone deacetylase; H3K4, Histone H3 lysine K4; M1 or M2, Macrophages; ncRNA, Non-coding RNA; lncRNA, Long non-coding RNA; circRNA, Circular RNA; miR, micro RNA, mRNA, messenger RNA, m^6^A, N6-methyladenosine; m^5^C, 5-methylcytosine; m^1^A, N1-methyladenosine.

**Figure 3 pharmaceuticals-16-00141-f003:**
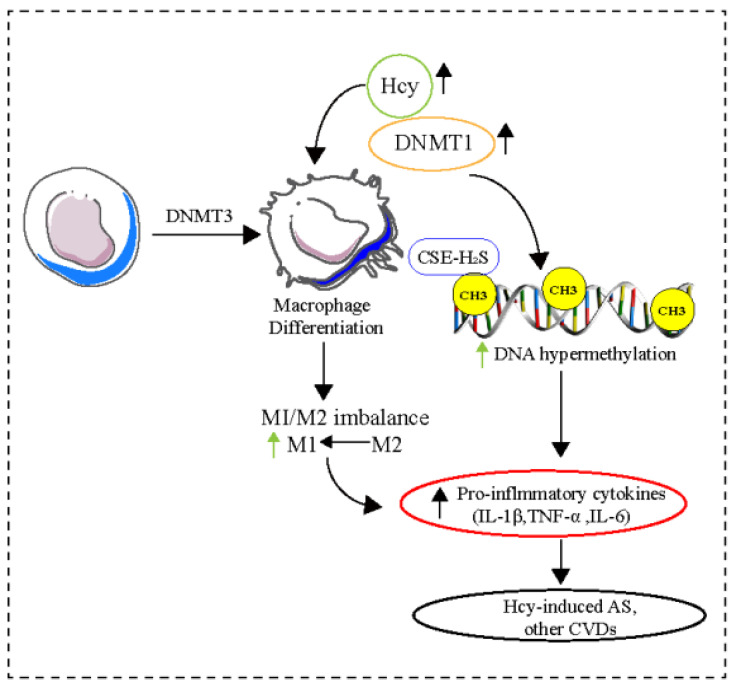
DNA methylation and macrophage polarization in CVDs. DNA hypermethylation and impaired CSE-H2S production in macrophages are associated with pro-inflammatory cytokines production and M1/M2 imbalance followed by various CVDs. Abbreviations: DNMTs, DNA methyltransferases; Hcy, Homocysteine; M1/M2, Macrophages; CSE-H_2_S, Cystathionine γ-lyase—hydrogen sulfide; AS, Atherosclerosis; IL, Interleukin; TNF-α, Tumor necrosis factor α; CVDs, Cardiovascular diseases.

**Figure 4 pharmaceuticals-16-00141-f004:**
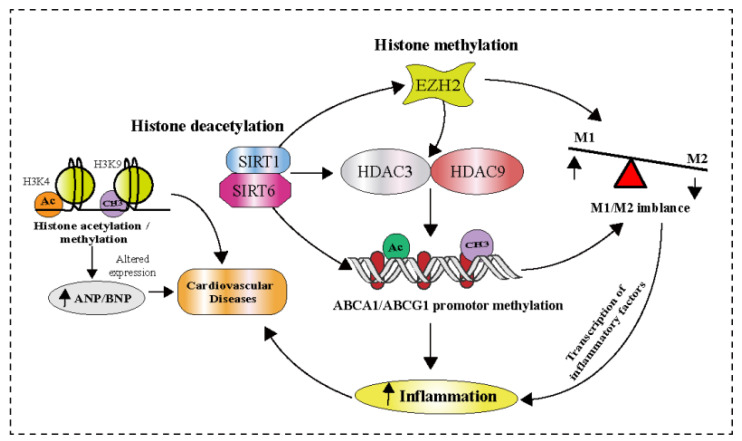
Histone modifications contribution to the regulation of macrophage polarization and inflammatory repertoire. Histone acetylation and demethylation are linked to the altered ANP and BNP gene expression in the myocardium. Further, EZH-2 exacerbates atherosclerosis and accelerates foam cell development by altering macrophage differentiation. Abbreviations: M1/M2, Macrophages; EZH-2, zeste homolog 2; HDAC, histone deacetylase; SIRT, Sirtuin; ANP, atrial natriuretic peptide; BNP, B-type natriuretic peptide; ABCA-1, ATP binding cassette subfamily A member 1; ABCG-1, ATP binding cassette subfamily G member.

**Table 1 pharmaceuticals-16-00141-t001:** Epigenetic regulation of macrophage polarization in different cardiovascular diseases.

Epigenetic Modifications	Enzyme or Targets	Macrophage Type	Function	CVDs	References
DNA methylation	DNMT1, DNA hydroxymethylation, CpG hypermethylation	M1/M2	Endothelial cell dysfunction, Macrophage differentiation	Atherosclerosis, Cardiac hypertrophy, Hcy-induced atherosclerosis	[51,52,53,54,55,56,57,58,59,60,61,62]
Histone modifications (Methylation/ Acetylation)	ANP, BNP, HDAC3, SIRT-6/1, EZH-2	M1/M2	Chromatin remodeling, Gene transcription	Myocardial remodeling, Carotid artery stenosis, Atherosclerosis	[63,64,65,66,67,68,69,70,71,72,73,74]
RNA regulation (m^6^A/m^5^C/m^1^A)	CPEB4, TTP, METTL3/14, YTHDF2/3, FTO, DNMT2, ICAM-1	M1/M2	Gene expression, Encoding pro-inflammatory and anti-inflammatory factors	Atherosclerosis, Acute coronary syndrome	[75,76,77,78,79,80,81,82,83,84,85,86,87]
Non-coding RNAs *miRNAs*	MAML1, NOTCH, SIRP α, KBTBD7, PDK1, NOX2, NF-κB	M1/M2	Gene regulation by inhibiting miRNAs or miRNA/RBP sponge transcription	Atherosclerosis, Myocardial infraction, Cardiac hypertrophy, Myocarditis, Heart failure	[88,89,90,91,92,93,94,95,96,97,98,99,100,101,102,103,104,105,106,107,108,109,110]
*lncRNAs*	API5	M1/M2	Epigenetic and transcriptional modifications, Chromatin remodeling, RNA splicing	Atherosclerosis, carotid artery disease	[111,112,113,114,115,116,117,118,119,120,121]
*circRNAs*	SP1, PARP	M1/M2	Gene regulation by inhibiting miRNAs or miRNA/RBP sponge transcription	Myocardial infarction, Ventricular dysfunction, Myocardial fibrosis	[122,123,124]

Epigenetic regulations are associated with altered gene expression and control a variety of cellular activities, including macrophage polarization and contribute to the pathophysiology of CVDs. Abbreviations: DNMT1, DNA methyltransferase 1; ANP, atrial natriuretic peptide; BNP, brain natriuretic peptide; HDAC3, histone deacetylase 3; SIRT-6, Sirtuin 6; EZH-2, Enhancer of zeste homolog 2; CPEB4, cytoplasmic polyadenylation element binding protein 4; TTP, tristetraprolin; Dnmt2, tRNA (cytosine(38)-C(5))-methyltransferase; MALAT, metastasis associated lung adenocarcinoma transcript 1; FTO, fat mass and obesity-associated protein; YTHDF, YTH N6-methyladenosine RNA binding protein; m^6^A, N6-methyladenosine; m^5^C, 5-methylcytosine; m^1^A, N1-methyladenosine; ICAM-1, Intercellular Adhesion Molecule 1; MAML1, mastermind-like protein 1; ABCA1, ATP-binding cassette transporterA1; PPAR-γ, peroxisome proliferator-activated receptor-γ; LXR, liver X receptor α; SIRP α, signal regulatory protein α; KBTBD7, BTB domain containing 7, PDK1, 3-phosphoinositide-dependent protein kinase 1; NOX2, NADPH oxidase 2, NOTCH, Notch homolog; API5, Apoptosis Inhibitor 5; NF-κB, nuclear factor kappa-light-chain-enhancer of activated B-cells; M1/M2: macrophages; RBPs, RNA-binding proteins.

**Table 2 pharmaceuticals-16-00141-t002:** ncRNAs association with macrophage polarization in the pathophysiology of cardiovascular diseases.

CVDs	miRNA	lncRNA	circRNA	Targets	Macrophage Regulatory Expression	References
Atherosclerosis	miR-133b-3p, miR-17-5p, miR-378a-3p, miR-let7, miR-205-5p,	lncRNA-MALAT1, lncRNA-MIAT, lncRNA- CERNA1, lncRNA-RAPIA	circCdyl	MAML1, NOTCH, ABCA1, SIRP α, NF-kB, API5	Increased M1expression and M1/M2 imbalance	[95,96,97,98,99,100,101,102,103,104,105,106,107,108,109,110]
Myocardial infarction	miR-21-5p, miR-375-3p, miR-204-5p, miR-148b-3p, miR-106b-5p	lncRNA-TCONS-00019715, lncRNA- Mirt2	-	KBTBD7, MAPK, p38/NF-kB, PDK1, NOX2, PPARγ	M1/M2 imbalance	[111,112,113,114,115,116,117,118,119,120,121]
Ischemia-reperfusion injury	miR-182	-	-	TLR4/NF-κB/PI3K/Akt	M1/M2 imbalance	[122,123,124]
Viral myocarditis	miR-155-5p, miR-21-5p, miR-146b-5p, miR-223	lncRNA-NRON, lncRNA-AK085865, lncRNA-MEG3	circCdyl	TLR, Pknox, TRAF6/ NF-κB, IRF4/ C/EBP-δ	M1/M2 imbalance	[124]

Studies show that the differential expression of specific ncRNAs in the myocardium alters macrophage polarization followed by M1/M2 imbalance and contributes to the pathophysiology of CVDs. Abbreviations: miRNA, microRNA; lncRNA, long non-coding RNAs; circRNA, circular RNA; MALAT1, metastasis-associated lung adenocarcinoma transcript 1; MIAT, myocardial infarction associated transcript; CERNA1, competing endogenous lncRNA 1 for miR-4707-5p and miR-4767; RAPIA, lncRNA associated with the progression and intervention of atherosclerosis; Mirt2, myocardial infarction-associated transcript 2; NRON, non-coding repressor of NFAT; MEG3, maternally expressed 3; M1/M2, macrophages; TRAF6, TNF receptor associated factor 6; MAML1, mastermind-like protein 1; ABCA1, ATP-binding cassette transporterA1; PPAR γ, peroxisome proliferator-activated receptor-γ; SIRP α, signal regulatory protein α; KBTBD7, BTB domain containing 7, PDK1, 3-phosphoinositide-dependent protein kinase 1; NOX2, NADPH oxidase 2, NOTCH, Notch homolog; API5, Apoptosis Inhibitor 5; NF-κB, nuclear factor kappa-light-chain-enhancer of activated B-cells; TLR, toll-like receptors; C/EBPδ, CCAAT/enhancer binding protein delta.

## Data Availability

Not applicable.

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
