# Peer review of "Epigenetic Regulation of Macrophage Polarization in Cardiovascular Diseases"

_pharmaceuticals, 2023, doi:10.3390/ph16020141_

Round 1

Reviewer 1 Report

Critique

Virtually every sentence is poorly written, the text reads like it was google-translated from some other language without further correction. Lack of basic knowledge of English in many cases alter meaning of statements that make you think that statements lack logics, or that authors cannot tell the difference between cause and effect. Some examples are given below. Manuscript will be substantially improved if authors used help from experts in English language, and if they give text to other colleagues that can pick up logical errors and redundant statements.

I was not able to evaluate the figures since they were not included with the manuscript. Supplementary files cannot be open on my Mac.

False statements:

Line 22: “Epigenetic modifications … regulate … DNA damage” is a false statement

Line 40: “CVDs … irreversibly alter the vascular architecture” is a false statement

Line 52: “cardiomyocyte-mediated injury” – not sure what does it mean

Line 57: “macrophages … are transformed into different functional phenotypes” is a false statement

Line 146: “In multicellular organisms” – do you want to say that there is no epigenetic regulation in unicellular organisms?

Line 150: “RNA modifications (e.g., mRNA and non-coding RNAs)” – false statement, mRNA and ncRNAs are NOT “RNA modifications”

Line 156: “DNA methylation is the main epigenetic modification” – false statement

Line 174: “atherosclerotic plaques … produce inflammatory cytokines” 

Line 196: “blood DNA hypermethylation” – it looks like authors cannot tell difference between “hypo” and “hyper”. Check ref 59

Line 203: “that specifically act on the amino acids arginine and lysine in the histone N-terminal tails” – false statement, there are other amino acid residues that undergo modifications

Needs to be clarified:

Line 59: M0 sub-group is excluded from the discussion, why?

Line 97: in this paragraph, authors use CCR- and CCR2- to describe apparently the same sub-population of macrophages. What are the differences between these two macrophage types?

Line 98: What is the relationship between CCR2 and M1/M2?

Line 107: “CCR2- macrophages originate from fetal monocyte progenitors and primitive yolk sac and” – this statement implies that this macrophage sub-population is produced only at the fetal stage, and then it persists throughout lifespan. Is it correct?

Line 106: “MHC-IIlowCCR2- and MHC-IIhighCCR2- are two resident cardiac macrophage subsets identified by genetic fate mapping under steady-state conditions, whereas MHC-IIhighCCR2+” – So what? Do these subpopulations reflect stochastic cell-to-cell differences, or they have distinct functions?

Line 112: “Although the complicated molecular mechanisms of polarization are yet unknown” – What polarization is it about?

Line 117: “cardiac macrophages are actively involved in … inflammation followed by myocardial injuries” Authors claim that inflammation is followed by injury (for example, myocardial infarction). Apparently, it is not about inflammation associated with/caused by the injury, but rather about chronic inflammation that precedes/contributes to the injury. Clarify

Line 137: “connexin 43 (Cx43)-containing gap junction protein” – how can a gap junction protein “contain” connexin 43??

Line 145: “heritable changes in the gene expression and function” – what do you mean by “gene function”?

Line 149: “These modifications” – What modifications?

Line 161: “is required for normal development” this statement implies that there is also abnormal development where epigenetics is not required. Clarify

Line 178: “followed by increased DNA methylation, and pro-inflammatory cytokines production” – DNA methylation is a repression mark, its increase therefore cannot be related to the cytokine production. Clarify

Line 192: “gene-specific promoter DNA methylation status of M1/M2 polarization markers significantly changed” – need to be more specific (i.e., what genes, what changes) because it is relevant to the topic

Line 196: “blood DNA hypermethylation” – what DNA is it about? 

Line 211: “histone H3 lysine K4 (H3K4) methylation” – be more specific (is it a global change per genome, or gene-specific)

Lines 216-218: So what?

English:

Remove most of “moreover”, “further”, “furthermore” from the text

Line 76: “In this review, we summarize the epigenetic regulations” – poor English

Line 83: “M1 macrophage phenotype … secretes pro-inflammatory cytokines” – poor English

Line 96: “… different subset of macrophages that can be differentiated through cell surface expression of C-C chemokine receptor type 2 (CCR2) and genetic lineage tracing” – how is it possible to differentiate cells through "genetic lineage tracing"?

Line 113: “the topic offers an opportunity to decipher the mechanisms underlying cardiac pathophysiology” – poor English

Line 128: “Dectin-1-polarized M1 macrophages, which are primarily expressed on cardiac macrophages” – this statement makes no sense

Line 147: “Epigenetic regulations” – poor English

Line 161: “The majority of DNA methylation” – poor English

Lack of logics:

Lines 67-78: this paragraph makes little sense

Lines 172-176: These two sentences make no sense in the context of this chapter

Lines 180-182: So what? How is it related to the epigenetics?

Lines 182-184: This sentence makes no sense

Lines 199-200: This sentence makes no sense

Line 215: “Moreover, in cell-specific histone methylation, the expression of accompanying lysine methyltransferases alters in carotid arteries” - This sentence makes no sense

Line 216: “significant differences in histone methylation modifications have been reported between offspring of hypercholesterolemic and normocholesterolemic mothers” – There is no such a thing as “histone methylation modifications”. What is “difference between offspring”? Do authors want to say that they homogenized the whole organism and measured histone modifications in the homogenate?!

Lines 218-222: This sentence makes no sense

Line 218: “A significantly low methylation level in H3K27 and H3K9 has been observed in CVD patients” – What does “significantly low” mean? There is no such a thing as “methylation level in patients". Also, this statement contradicts the second part of the same sentence. 

Redundances:

There are many redundant statements (about M1/M2 phenotypes, polarization, inflammation as a cause of CVD, etc) in the text that need to be removed

Author Response

Thank you. We improve the manuscript according to the respectable reviewer's comments and suggestions following by article clarity and readability.  To improve the manuscript clarity and readability “Professional English and Academic Editing Services–Editage” has provided us assistance (kindly find the attachment). After amenable changes in the manuscript writing, we are ready to submit this revised version of the manuscript.

Figures (Figure-1, Figure-2, Figure-3 and Figure -4) and Tables (Table-1 and Tbale-2), are added as in the main text as well. After amenable changes in the manuscript, we are ready to submit this revised version of the manuscript.  The revised versions of the manuscript and changes to the manuscript are indicated by blue highlights. I believe that we have undertaken substantial changes, and we await the results of the review of this revised version of the manuscript.

Reviewer 2 Report

Dear Authors,

in my opinion references shoud be up-to-date and text should be checked by English Editing

Other things are ok.

Author Response

Thank you. We improve the manuscript according to the respectable reviewer's comments and suggestions following by article clarity and readability.  To improve the manuscript clarity and readability “Professional English and Academic Editing Services–Editage” has provided us assistance (kindly find the attachment). After amenable changes in the manuscript writing, we are ready to submit this revised version of the manuscript.

Figures (Figure-1, Figure-2, Figure-3 and Figure -4) and Tables (Table-1 and Tbale-2), are added as in the main text as well. After amenable changes in the manuscript, we are ready to submit this revised version of the manuscript.  The revised versions of the manuscript and changes to the manuscript are indicated by blue highlights. I believe that we have undertaken substantial changes, and we await the results of the review of this revised version of the manuscript. Thank you

Reviewer 3 Report

The manuscript is of merit and well written. I would just suggest to summarize the content with an illustration.

Author Response

Thank you. We improve the manuscript according to the respectable reviewer's comments and suggestions following by article clarity and readability.  To improve the manuscript clarity and readability “Professional English and Academic Editing Services–Editage” has provided us assistance (kindly find the attachment). After amenable changes in the manuscript writing, we are ready to submit this revised version of the manuscript.

Figures (Figure-1, Figure-2, Figure-3 and Figure -4) and Tables (Table-1 and Tbale-2), are added as in the main text as well. After amenable changes in the manuscript, we are ready to submit this revised version of the manuscript.  The revised versions of the manuscript and changes to the manuscript are indicated by blue highlights. I believe that we have undertaken substantial changes, and we await the results of the review of this revised version of the manuscript.

Thank you for your consideration.

Round 2

Reviewer 1 Report

The manuscript reads better, some edits are needed, please see below

Line 47: “…into the genetic and epigenetic factors underlying CVDs [8,9]. Epigenetic factors

have also been reported to be involved in the pathophysiology of CVDs [10]. In particular, epigenetic modifications, … have recently been linked to … CVDs” – Redundant sentences

Line 49: “epigenetic modifications, i.e., heritable changes in gene expression without altering

the primary DNA sequence” – epigenetic modifications refer to chemical modifications of DNA or histones that are associated with changes in gene expression

Line 78: “Moreover, macrophage polarization is closely related to the regulation of inflammatory pathways and the modulation of epigenetic mechanisms” – redundant, remove

Line 88: “The M1 macrophage phenotype is activated by LPS, IFN-γ” – redundant with the above section

Line 187: “DNA hypermethylation, which may trigger the elevation of pro-inflammatory cytokines production in macrophages” – this statement contradicts the established role of DNA methylation (in the promoter regions) in gene silencing

Line 209: “Histone proteins are the major regulatory components of chromatin” – on the contrary, NHGRI defines histones as “proteins that provide structural support for a chromosome”

Line 213: “primarily act on the N-terminal tail” – this sentence refers to the tail of one particular protein, which one?

Line 218: “Enhanced histone acetylation and dimethylation” – be more specific, which residues? Is it “di” or “de”?

Line 220: “whereas Sarcoplasmic reticulum Ca(2+) ATPase gene (SeRCA2a) expression does not differ between LV and right ventricle (RV)” – remove

Line 221: “Additionally, there was…” – what does it have to do with macrophages?

Line 224: “Studies have shown…” – in this paragraph remove all sentences that are not relevant to macrophages 

Line 275: “RNA-binding proteins” – how is it relevant to RNA modification? Nobody calls RNA-binding proteins “readers”

Line 282: “nitrogenous bases” – what is it?

Line 291: “functional protein regulation” – what does it have to do with m6A?

Line 291: “m6A RNA modifications…” – remove this sentence

Line 327: “The non-coding genome is essential for genetic programming” – what does it mean?

Line 417: In this paragraph, remove all sentences that are not relevant to macrophages

Line 450: “signal-stranded” – what is it?

Line 458: “Further, m6A modification in hsa_circ_0029589 reduces hsa_circ_0029589 expression” – what does this sentence mean?

Line 467: “NGS analyses of heart tissue” – was it genome sequencing? How is it relevant to macrophages? In this paragraph, remove sentences not relevant to macrophages

Line 473: “specific protein 1 (SP1)” – it is “specificity protein 1”

Line 531: “This demonstrates that synthetic medicines that precisely target epigenetic mechanisms can be effective immunomodulatory therapeutics” – this conclusion has no connection to the previous sentence about Myh7 gene

Line 532: “An example…” – an example of what? What does it have to do with macrophage epigenetics?

Author Response

Thank you. We improve the manuscript according to the respectable reviewer comments and suggestions following by article clarity and readability.  
